# Generalizing Experience for Language Agents with Hierarchical MetaFlows

**Shengda Fan**[1], **Xin Cong**[2*], **Zhong Zhang**[3], **Yuepeng Fu**[3], **Yesai Wu**[3]
**Hao Wang, Xinyu Zhang, Enrui Hu, Yankai Lin**[1*]
[1]Gaoling School of Artificial Intelligence, Renmin University of China
[2]Department of Statistics and Data Science, Tsinghua University
[3]Department of Computer Science and Technology, Tsinghua University
{fanshengda,yankailin}@ruc.edu.cn
congxin1995@mail.tsinghua.edu.cn

## Abstract

Recent efforts to employ large language models (LLMs) as agents have demonstrated promising results in a wide range of multi-step agent tasks. However, existing agents lack an effective experience reuse approach to leverage historical completed tasks. In this paper, we propose a novel experience reuse framework MetaFlowLLM, which constructs a hierarchical experience tree from historically completed tasks. Each node in this experience tree is presented as a MetaFlow which contains static execution workflow and subtask required by agents to complete dynamically. Then, we propose a Hierarchical MetaFlow Merging algorithm to construct the hierarchical experience tree. When accomplishing a new task, MetaFlowLLM can first retrieve the most relevant MetaFlow node from the experience tree and then execute it accordingly. To effectively generate valid MetaFlows from historical data, we further propose a reinforcement learning pipeline to train the MetaFlowGen. Extensive experimental results on AppWorld and WorkBench demonstrate that integrating with MetaFlowLLM, existing agents (e.g., ReAct, Reflexion) can gain substantial performance improvement with reducing execution costs. Notably, MetaFlowLLM achieves an average success rate improvement of **32.3**% on AppWorld and **6.2**% on WorkBench, respectively. The code is available at https://github.com/RUCBM/MetaFlowLLM.

## 1 Introduction

Large Language Models (LLMs) have demonstrated remarkable capabilities in a wide range of multi-step agent tasks, including interactive coding [39, 32], web browsing [40, 46], and embodied housework [30, 20]. Existing work leveraging LLMs as autonomous agents that can iteratively take actions based on the execution feedback, enabling dynamic adaptation throughout the task execution process [41, 29]. Despite its effectiveness, when handling a large volume of tasks, earlier work tends to deal with each task independently, lacking the ability to summarize prior completed tasks to accumulate experience and enhance future task execution. As a result, the inability to learn from prior experiences results in the repetition of unnecessary steps (e.g., redundant trial-and-error) even when performing similar or identical tasks, thereby leading to ineffective and inefficient execution.

To address this limitation, two primary approaches are commonly employed: (1) **Trajectory-based Experience**, which involves constructing a database of historical task completion trajectories and retrieving the most relevant examples as in-context demonstrations to assist in executing new tasks [3, 45]; and (2) **Guideline-based Experience**, which involves summarizing prior experiences into

---
*Corresponding author.

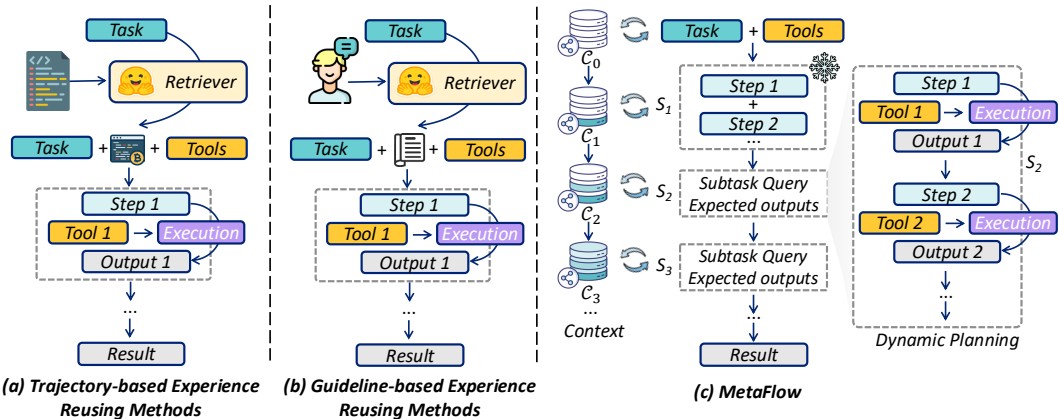

Figure 1: Illustration of three experience reuse approaches.

abstract natural language guidelines and using these guidelines to inform the execution of future tasks [44, 12]. However, trajectory-based experience often introduces fine-grained but task-irrelevant details, as even ostensibly similar tasks may exhibit divergent execution pathways, leading to reduced robustness when task contexts or environmental conditions change. Conversely, guideline-based experience typically presents experience in a coarse-grained form, which may omit critical procedural nuances, lacking the precision, structure, and robustness needed for reliable execution.

Humans do not learn experience by directly reusing specific procedures or merely summarizing abstract guidelines. A more effective approach is to construct an experience hierarchy based on the similarity between tasks: the more similar the tasks are, the more concrete procedures can be directly reused; the less similar they are, the more dynamic execution is needed [4, 7]. For instance, consider the following two tasks: (1) *Send an email to attendees of the November 30 event with the title 'Remember to attend this event.'* (2) *Remind attendees of the December 1 event via email titled 'Remember to attend this event.'* As humans, we can recognize that many steps in these two tasks can be reused, including searching for the event using `calendar.search_events`, extracting participant emails, and sending emails via the `email.send_email` tool. For those differences (e.g., extracting the event date from the user query), adjustments can be made based on the specifics of each task. Motivated by this human ability, we propose **MetaFlowLLM**, a novel framework that builds a hierarchical experience tree from historical task execution data to improve the effectiveness and efficiency of LLMs. Specifically, as illustrated in panel (c) of Figure 1, each node in the experience tree represents an abstracted workflow named **MetaFlow**, which contains both concrete execution workflow (i.e., the static step) and subtasks (i.e., the dynamic step) that need LLMs to execute dynamically. The closer to the leaf nodes, the more static steps are in the MetaFlow; the closer to the root node, the more dynamic steps it contains. When confronted with a new task, MetaFlowLLM retrieves from the hierarchical experience tree to identify the most appropriate MetaFlow node and initiates its execution. For components corresponding to concrete workflows, it executes them directly. For components requiring dynamic execution, an autonomous agent is responsible for interpreting the subtask description and completing it accordingly.

Implementing MetaFlowLLM presents two key challenges: (1) how to abstract MetaFlows given arbitrary two task execution trajectories: considering that manually summarizing MetaFlows is both time-consuming and labor-intensive, and that directly prompting powerful LLMs such as GPT-4o or DeepSeek R1 does not yield satisfactory results, we propose a reinforcement learning pipeline based on verifiable rewards to train an effective MetaFlow generator, called MetaFlowGen. Specifically, we introduce two distinct rewards: *i*) a **correctness reward**, which quantifies the proportion of queries for which the generated MetaFlow yields correct results, and *ii*) an **efficiency reward**, which incentivizes the preservation of static steps while minimizing unnecessary invocations of LLMs. Leveraging these rewards, we initially construct a high-quality supervised fine-tuning dataset via rejection sampling to enable effective cold-start. Subsequently, we directly optimize the MetaFlow generator using reinforcement learning to maximize the defined rewards. (2) how to construct an experience tree from existing historical data: we draw inspiration from hierarchical clustering techniques [24, 25] and propose a Hierarchical MetaFlow Merging algorithm. This algorithm iteratively merges the most

similar nodes using the MetaFlowGen until the tree is fully constructed. To ensure the usability of the experience tree, we incorporate a pruning algorithm based on the reward of the merged nodes, which helps maintain its efficiency and effectiveness.

To validate the effectiveness of our proposed method, we conduct extensive experiments on two distinct scenarios including AppWorld [32] and WorkBench [31]. Experimental results demonstrate that integrating MetaFlow into existing LLM-based agents leads to substantial performance improvements while simultaneously reducing execution costs. Notably, MetaFlowLLM achieves an average improvement of **32.3**% on AppWorld and **6.2**% on WorkBench, surpassing both trajectory-based and guideline-based experience reuse approaches.

Our contributions are as follows:

- We introduce MetaFlowLLM as a novel mechanism for experience reuse in LLM-based agents, enabling them to generalize structured workflows across similar tasks.
- We propose a reinforcement learning-based training pipeline for the MetaFlow generator to improve the MetaFlow generation capabilities of LLMs.
- Through comprehensive experiments on AppWorld [32] and WorkBench [31], we demonstrate that our method produces high-quality workflows that significantly improve LLM performance and reduce computational overhead.

## 2 Related Work

**LLM-based Agents**   With advances in the instruction-following and reasoning capabilities of LLMs [1, 17, 38, 13], their applications have expanded beyond traditional natural language processing tasks such as question answering [27] and information extraction [10, 9]. Researchers have begun to use LLMs as autonomous agents for tasks such as code generation [26, 32], travel planning [37], web browsing [40, 46], and embodied housework [30, 20]. Currently, the two most representative LLM agent frameworks are ReAct [41] and Reflexion [29]. However, these two vanilla agents are incapable of utilizing historical data to form experience to promote future tasks. To address this limitation, we propose the MetaFlowLLM framework, which enables agents to accumulate, retrieve, and leverage past experiences to enhance reasoning, decision-making, and generalization across tasks.

**Experience Reuse in LLM-based Agents**   Current approaches to experience reuse in LLM agents primarily follow two paradigms: (1) trajectory-based experience: direct utilization of complete or abstracted trajectories as contextual examples [3, 45, 34], which often suffer from overfitting to specific instances, and (2) guideline-based experience: manual or LLM-generated summarization of experiences into natural language instructions that are incorporated into the agent's prompt [44, 12]. For the former one, they retain excessive task-specific details and not sufficiently generalizable. For the latter one, the abstract guideline in natural language may lacking the precision, structure, and robustness needed for reliable execution. To address these limitations, we propose a novel experience reuse framework, MetaFlowLLM, that constructs a hierarchical experience tree covering experiences at different levels. Compared to AWM [34], which extracts reusable workflows from historical trajectories and still relies on prompt-based experience reuse driven by the LLM's reasoning capability, MetaFlowLLM explicitly models reusable workflows as structured graphs with static and dynamic nodes, and further optimizes their granularity through reinforcement learning.

**Workflow Generation**   Process Automation [5] has long been a key goal in technology, aiming to automate repetitive tasks and improve operational efficiency. Traditional Robotic Process Automation relies on manually annotated workflows to abstract such tasks [19, 15, 35, 2, 11]. With the advent of LLMs, many efforts have focused on leveraging their capabilities to replace manual labor in workflow generation [42, 43, 16, 36, 21]. However, generating large-scale workflows with complex logical structures based solely on queries and available tool lists remains a significant challenge for current LLMs [8, 42], primarily due to the lack of intermediate execution feedback. To address this limitation, our work takes a more pragmatic approach by refraining from generating all specific execution steps. Instead, we abstract similar workflows into a unified MetaFlow, where shared steps are fixed and sample-specific components are delegated to sub-tasks handled by dynamic agents. In this way, MetaFlowLLM provides a balanced framework for automation—capturing common workflow patterns while maintaining the flexibility to adapt to diverse tasks.

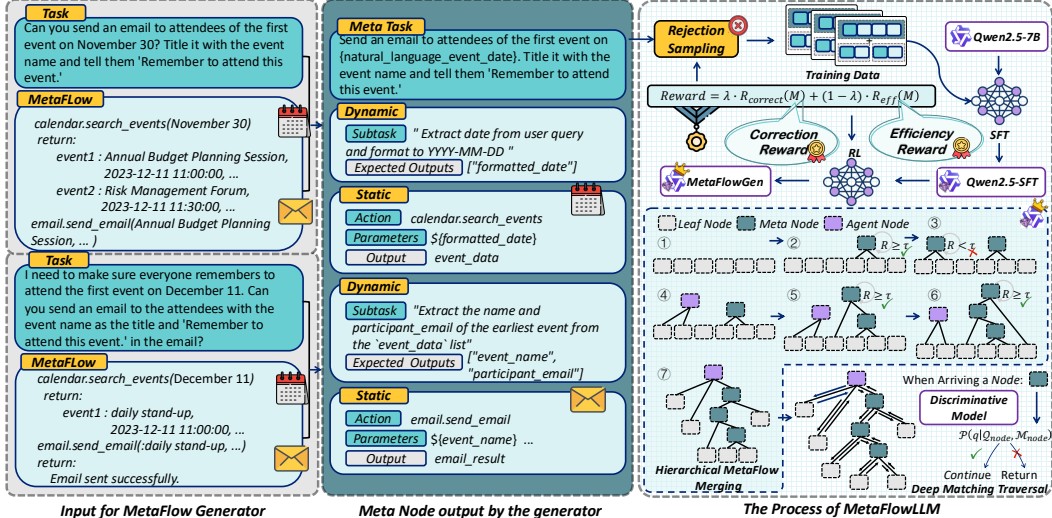

Figure 2: Illustration of the MetaFlowLLM framework. On the left, the task and corresponding MetaFlow (since the example in this figure is from the leaf node, it is referred to as a trajectory) serve as inputs to the generator. The middle section depicts the meta node (comprising the meta task and MetaFlow) generated by the MetaFlow generator. On the right, the process of MetaFlowLLM framework is outlined, including: 1) the training process, where a rejection sampling approach constructs the cold-start dataset, followed by SFT and RL to train the MetaFlowGen; and 2) the deployment process, which involves constructing the hierarchical experience tree and performing deep matching traversal to find the most appropriate MetaFlow node.

## 3 Methodology

In this section, we introduce the MetaFlowLLM framework, which constructs a hierarchical experience tree containing different level MetaFlows to leverage historical completed tasks. We first formally define the MetaFlow structure and its operational rules in Section 3.1. Then, we propose a reinforcement learning pipeline to derive a dedicated MetaFlowGen to generate MetaFlows, as detailed in Section 3.2. Finally, we present the hierarchical experience tree construction and deep matching traversal algorithms in Section 3.3.

### 3.1 MetaFlow Definition

As illustrated in part (c) of Figure 1, MetaFlow is a hybrid execution framework composed of static procedural steps and dynamic LLM-based agent steps, all operating over a shared and mutable context $\mathcal{C}$. Formally, a MetaFlow $\mathcal{M} = \{s_1, s_2, \ldots, s_n\}$ is defined as a sequence of steps, each of which can be either a static step $s_i^S$ or a dynamic step $s_i^D$. The shared context $\mathcal{C}$ is initialized with the user task. As the MetaFlow progresses, $\mathcal{C}$ is iteratively updated by both static steps and the outputs of online planning agents. In practice, $\mathcal{C}$ may be an implicit variable scope (e.g., in Python), or an explicit key-value store tracking agent decisions and tool outputs.

At runtime, given a task $q$, MetaFlowLLM executes each step sequentially in the order defined by $\mathcal{M}$. For each step $s_i$, the framework checks the step type:

- If $s_i$ is a static step, consisting of operations such as tool calls or code execution, it is directly executed, and the context is subsequently updated as $\mathcal{C}_i = s_i^S(\mathcal{C}_{i-1})$.
- If $s_i$ is a dynamic agent step, MetaFlowLLM adapts the subtask of $s_i$ based on the task $q$ and the current context $\mathcal{C}_{i-1}$, producing a state-specific prompt $p_i$ as subtask prompt. $p_i$ is then used to initialize an LLM-based agent $A$, such as ReAct [41] or Reflexion [29]. The LLM agent $A$ interacts with the environment over multiple turns, each comprising an action and the subsequent observation. Formally, the agent $A$ produces an action sequence $\{a_i^1, a_i^2, \ldots, a_i^k\} = A(p_i)$, where each action $a_i^j$ is executed and the context is updated accordingly:

$$\mathcal{C}_i^j = s_i^D(\mathcal{C}_i^{j-1}, a_i^j), \quad \mathcal{C}_i^0 = \mathcal{C}_{i-1}.$$

The process continues until the agent determines that the subtask is complete. The final context after executing all actions is denoted as $C_i = C_i^k$.

## 3.2 Training MetaFlowGen with Reinforcement Learning

To generate effective MetaFlows, we adopt a reinforcement learning pipeline to train a MetaFlow generator named MetaFlowGen. As illustrated by the right part of Figure 2, the pipeline follows a two-stage approach: supervised fine-tuning (SFT) to initialize the model for cold-start, and rule-based reinforcement learning using GRPO [28] to enhance the MetaFlow generation capability.

As illustrated in the left part of Figure 2, the MetaFlow generator is designed to take two samples as input: each consisting of tasks with the corresponding MetaFlows and the relevant tool documentation. The goal is to generate a high-level MetaFlow $\mathcal{M}$ that captures their common patterns through static steps and handles their different procedures using dynamic agents. Formally, the MetaFlow generation process can be expressed as:

$$\mathcal{M}, \mathcal{Q} = \pi_\theta \left( q_1, q_2, m_1, m_2, \mathcal{A} \right), \tag{1}$$

where $\theta$ is the parameter of the MetaFlowGen, $q_1, q_2$ are the input tasks respectively, $m_1, m_2$ are their corresponding MetaFlows, and the tool documentation $\mathcal{A}$ provides necessary background knowledge. As illustrated in the middle part of Figure 2. the output of the MetaFlowGen consists of the MetaFlow $\mathcal{M}$ and its associated meta task $\mathcal{Q}$.

**Supervised Fine-Tuning for Cold-Start.** Following previous work [14], we use SFT as a cold-start method. The SFT training dataset, denoted as $D_{\text{SFT}} = \{(q_1, q_2, m_1, m_2, \mathcal{M}, \mathcal{Q})^{(i)}\}_{i=1}^{|D_{\text{SFT}}|}$, is constructed through rejection sampling [6]. Specifically, we maintain a list of golden nodes, initialized with the samples from the training set (referred to as leaf nodes). In each iteration, we sample two golden nodes and pair them with manually crafted in-context learning samples to form an input prompt. This prompt is then used to generate the corresponding meta task and MetaFlow via an LLM as a MetaFlow generator. We execute this MetaFlow on all its leaf nodes, and if it passes all the test cases associated with these nodes (i.e., its correctness reward equals 1.0 defined in Equation 3), it is deemed correct and added to the golden nodes list. This iterative process continues until the dataset reaches the desired scale. It effectively expands both the dataset size and the complexity of the hierarchical structures, facilitating the subsequent construction of the hierarchical experience tree.

Formally, the objective function for supervised fine-tuning is defined as:

$$\mathcal{L}_{\text{SFT}}(\theta) = -\mathbb{E}_{(q_1, q_2, m_1, m_2, \mathcal{M}, \mathcal{Q}) \sim D_{\text{SFT}}} \left[ \log \pi_\theta(\mathcal{M}, \mathcal{Q} \mid q_1, q_2, m_1, m_2, \mathcal{A}) \right]. \tag{2}$$

**Rule-based Reinforcement Learning.** After completing the SFT phase, we further enhance the model using the reinforcement learning approach. In this phase, the training dataset is constructed by sampling from the list of golden nodes produced during SFT. To guide the learning process, we design a reward function that integrates two key objectives: correctness and efficiency.

- **Correctness Reward.** This term evaluates whether initializing the MetaFlow correctly solves the leaf nodes' tasks. The correctness reward is defined as the ratio of correctly solved leaf nodes:

$$R_{\text{correct}}(\mathcal{M}) = \frac{1}{N} \sum_{i=1}^{N} \mathbb{I}(\texttt{execute}(q_i, \mathcal{M}) = \texttt{expected}_i), \tag{3}$$

where $N$ is the number of leaf nodes, and $\mathbb{I}$ is the indicator function that checks whether the generated result matches the expected output.

- **Efficiency Reward.** This component evaluates the preservation of static actions and runtime efficiency. The efficiency reward is defined as the ratio of static operations to the minimum number of static operations between the two input workflows:

$$R_{\text{eff}}(\mathcal{M}) = \frac{\texttt{NumStaticOps}(\mathcal{M})}{\min(\texttt{NumStaticOps}(m_1), \texttt{NumStaticOps}(m_2))}, \tag{4}$$

This encourages workflows that maximize static step reuse and minimize redundant dynamic calls, improving both reusability and efficiency.

**Algorithm 1** Hierarchical MetaFlow Merging

---

**Require:** Leaf nodes $\mathcal{D} = \{(q_i, m_i)\}$, distance metric $d(\cdot, \cdot)$, MetaFlowGen $\theta$, correctness threshold $\tau$
**Ensure:** Hierarchical experience Tree $\mathcal{T}$
1: **Initialize:** Initialize the set of nodes $\mathcal{N} \leftarrow \mathcal{D}$
2: **while** $|\mathcal{N}| > 1$ **do**
3:     **Select the most similar node pair:** From $\mathcal{N}$, select two most similar nodes $n_1, n_2$ based on the distance metric $d(\cdot, \cdot)$
4:     **Generate candidate MetaFlow:** Merge $n_1$ and $n_2$ using the MetaFlowGen $\theta$ to produce a candidate MetaFlow $\mathcal{M}$
5:     **Evaluate its correctness:** Compute the correctness reward $R_{\text{correct}}$ for $\mathcal{M}$
6:     **if** $R_{\text{correct}} >= \tau$ **then**
7:         Use $\mathcal{M}$ as the new parent node to merge $n_1$ and $n_2$
8:         Remove $n_1, n_2$ from $\mathcal{N}$ and add $\mathcal{M}$ to $\mathcal{N}$
9:     **else**
10:        Use pure-agent node as the new parent node to merge $n_1$ and $n_2$
11:        Remove $n_1, n_2$ from $\mathcal{N}$
12:     **end if**
13: **end while**
14: **Final merge:** If there are more than one pure-agent nodes, merge them into a single top-level pure-agent node
15: **return** The complete experience Tree $\mathcal{T}$

---

The total reward is the weighted sum of the above two rewards[2], guiding the generator to produce MetaFlows that are both accurate and efficient.

$$R(\mathcal{M}, \mathcal{Q}) = \lambda \cdot R_{\text{correct}}(\mathcal{M}) + (1 - \lambda) \cdot R_{\text{eff}}(\mathcal{M}), \tag{5}$$

where $\lambda$ is a scaling factor for controlling the trade-off between accuracy and efficiency.

### 3.3 Hierarchical Experience Tree Construction and Utilization

For efficient inference, we begin by constructing a hierarchical experience tree from the leaf nodes in the historical completed tasks. This approach is detailed in Algorithm 1. As shown in the right part of Figure 2, the set of nodes to be merged, denoted as $\mathcal{N}$, is initialized as the set of leaf nodes $\mathcal{D}$. $\mathcal{N}$ is progressively merged and updated based on semantic similarity. Each merger results in an intermediate meta node $\mathcal{M}$. To ensure the effectiveness of the experience tree, we calculate the correctness reward of $\mathcal{M}$ using Equation 3. If the reward exceeds a threshold $\tau$, the meta node is added to $\mathcal{N}$. Otherwise, the two nodes are directly merged into a pure agent node.

At inference time, the goal is to efficiently identify the most appropriate MetaFlow to solve the task. As shown in Algorithm 2 and Figure 2, the deep matching traversal process begins by matching the test task $q_{\text{test}}$ with the experience tree $\mathcal{T}$, starting from the root node and moving downwards. The node-level matching is performed using a discriminative model $\mathcal{P}(q|\mathcal{Q}, \mathcal{M})$, which prompts an LLM to determine whether the task $q$ can be classified as an instance of a $\mathcal{Q}$ and can be solved by the $\mathcal{M}$. The search continues recursively through child nodes until the deepest matching node is found.

## 4 Experiments

### 4.1 Experimental Settings

**Datasets and Metrics.** We conduct experiments on two representative agent datasets: AppWorld [31] and WorkBench [32]. AppWorld serves as a representative dataset for interactive coding tasks, while WorkBench serves as a representative dataset for tool learning tasks in workplace settings. The evaluation metric in AppWorld is the **Task Goal Completion (TGC) Rate**, which is defined as the percentage of tasks where the agent passes all evaluation tests. We use two evaluation metrics in WorkBench: (1) **Accuracy**, which is defined as the percentage of tasks where the outcome from the agent's actions matches the expected outcome of the ground truth; and (2) **Side Effects**, which quantitatively evaluate unintended changes caused by tool calls. We conduct experiments under two

---

[2]We also incorporate a format-based penalty [14] in the reward design: a reward of $-1$ is assigned when the $\mathcal{M}$ fails to meet the required format constraints.

Table 1: TGC rate comparison of various models with different methods on AppWorld(%).

| Model | Agent Type | Configuration | | | |
|---|---|---|---|---|---|
| | | Base | *w/ Traj.* | *w/ Guideline* | *w/ MetaFLow* |
| Qwen2.5-7B | Reflexion | 7.6 | 32.7 | 13.5 | **37.4** |
| | ReAct | 6.4 | 37.4 | 17.0 | **43.9** |
| Qwen2.5-32B | Reflexion | 12.9 | 33.9 | 22.8 | **43.9** |
| | ReAct | 22.2 | 49.1 | 38.6 | **50.9** |
| GPT-4o-mini | Reflexion | 13.5 | 37.4 | 20.5 | **45.6** |
| | ReAct | 7.0 | 22.2 | 7.0 | **35.7** |

conditions: one where only the required tools are provided to LLMs, and another where all the tools are available. The dataset statistics can be found in Appendix D.

**Baselines.** To evaluate the performance of MetaFlow, we consider four base LLMs during the inference stage: two state-of-the-art open-source models, Qwen2.5-7B-Instruct [38] and Qwen2.5-32B-Instruct, and two proprietary models, GPT-4o-mini [18] and GPT-4o. In order to assess the generalizability of MetaFlow, we consider two agent frameworks as the backbone for dynamic nodes: (1) ReAct [41], which combines reasoning and acting, and (2) Reflexion [29], which enables the agent to leverage environmental feedback to reflect on mistakes and perform subsequent actions. We consider the following three baselines: (1) LLM agent without experience reuse, (2) Trajectory-based Experience reuse method (Denoted as *Traj.*), and (3) Guideline-based Experience reuse method (Denoted as *Guideline*). Further implementation details of the trajectory-based experience reuse and guideline-based experience methods can be found in Appendix E.

**Implementation Details.** We set $\tau$ to $1.0$ in all experiments to ensure the quality of the experience tree. The value of $\lambda$ is set to $0.7$. We use the all-MiniLM-L6-v2[3] model to encode the task, and the cosine similarity of the task embeddings is utilized as the distance metric.

In the SFT stage, we fine-tune the MetaFlowGen on Qwen2.5-7B-Instruct for 3 epochs using the AdamW optimizer [23] and a linear learning rate scheduler with a peak learning rate of $2 \times 10^{-5}$. Each mini-batch contains 32 examples, and the maximum sequence length is set as $8,192$ tokens. In RL stage, we adopt TRL [33] as our training framework. We set the training epochs to 2, batch size to 28, learning rate to $1 \times 10^{-6}$, KL coefficient to 0 [22], rollout number to 14. All experiments are conducted on 8 NVIDIA A800 40G GPUs.

## 4.2 Main Results

**Results on AppWorld.** Following [32], we evaluate the performance of LLMs using the ReAct and Reflexion prompting methods. The results are placed in Table 1, from which we derive that:

1. The improvement of the guideline methods on the challenging AppWorld is significantly smaller than that of trajectory-based methods. This suggests that, when faced with difficult tasks, the natural language guideline method may omit critical procedural details, lacking the precision, structure, and robustness required for reliable execution.
2. The proposed MetaFlowLLM method can effectively adapt to both the Reflexion and ReAct agent frameworks, leading to substantial performance improvements. Notably, MetaFlow increases TGC by **33.1**% and **31.6**% compared to the base Reflexion and ReAct agents, consistently outperforming the trajectory-based and guideline-based experience reuse methods. The results demonstrate the effectiveness of breaking down complex tasks into distinct sub-tasks, with clear and reusable transition logic between them. This decomposition allows to reuse static code and agent's plan from historically similar tasks, thereby increasing both execution stability and reliability.

**Results on WorkBench.** Following [31], we assess the performance of LLM agents using the ReAct [41] framework. Since the WorkBench dataset lacks pre-defined "golden" trajectories, we

---

[3]`https://huggingface.co/sentence-transformers/all-MiniLM-L6-v2`

Table 2: Accuracy and side effects comparison of various models with different methods on Work-Bench(%). The best results are marked in **bold** and the second-best results are marked with underline. Abbreviations are defined as follows. CRM: Customer Relationship Manager, PM: Project Management, MD: Multi-Domain.

| Model | Analytics | | Calendar | | CRM | | Email | | PM | | MD | | Avg | |
|---|---|---|---|---|---|---|---|---|---|---|---|---|---|---|
| | %acc↑ | %se↓ | %acc↑ | %se↓ | %acc↑ | %se↓ | %acc↑ | %se↓ | %acc↑ | %se↓ | %acc↑ | %se↓ | %acc↑ | %se↓ |
| GPT-4o | 30.8 | 53.8 | 53.0 | 19.7 | 30.9 | 14.5 | 32.8 | 25.9 | 8.16 | **0.0** | 16.3 | 48.9 | 27.2 | 33.1 |
| *w/ Traj.* | **48.5** | 47.4 | 48.5 | **10.6** | 54.5 | **10.9** | 41.3 | 25.9 | 20.4 | 4.13 | **25.1** | 55.1 | 35.3 | 32.7 |
| *w/ Guideline* | 34.6 | 44.9 | 53.0 | 21.2 | 40.0 | 23.6 | 31.0 | 27.6 | 8.16 | **0.0** | 15.6 | 52.4 | 28.5 | 34.2 |
| *w/ MetaFlow* | 44.9 | **37.2** | **63.7** | 12.1 | **60.0** | 14.5 | **46.6** | **24.1** | **24.5** | 6.12 | 24.5 | **40.8** | **40.8** | **26.9** |
| Qwen2.5-7B | 18.2 | 60.3 | 27.6 | **35.5** | 16.0 | 30.5 | 9.31 | **43.8** | 8.57 | **6.12** | 9.39 | 53.9 | 14.3 | **43.0** |
| *w/ Traj.* | **30.8** | 46.2 | 18.2 | 48.5 | 18.2 | 32.7 | 15.5 | 46.6 | 14.3 | 12.2 | 8.16 | 57.1 | 16.3 | 44.8 |
| *w/ Guideline* | 21.8 | 51.3 | **37.9** | 47.0 | 20.0 | 52.7 | 15.5 | 55.2 | 10.2 | 6.12 | 8.16 | 60.5 | **17.4** | 49.5 |
| *w/ MetaFlow* | 26.9 | 43.6 | 25.8 | 50.0 | 23.6 | 34.5 | 10.3 | 55.2 | 14.3 | 18.4 | 9.52 | 57.8 | 17.2 | 46.8 |
| Qwen2.5-32B | 6.41 | 88.5 | 57.6 | 24.2 | 12.7 | 43.6 | 27.6 | 24.1 | 6.12 | **0.0** | 12.2 | 61.9 | 19.2 | 47.2 |
| *w/ Traj.* | 25.6 | 70.5 | 37.9 | 40.9 | 41.8 | 20.0 | 8.62 | 58.6 | 20.4 | 12.2 | 15.0 | 60.5 | 23.2 | 49.0 |
| *w/ Guideline* | 19.2 | 79.5 | 59.1 | 21.2 | 27.3 | 23.6 | 31.0 | 36.2 | 10.2 | 2.0 | 15.0 | 57.8 | 25.2 | 43.3 |
| *w/ MetaFlow* | 24.4 | 69.2 | 59.1 | **19.7** | 27.3 | 23.6 | 37.9 | 32.8 | 12.2 | 4.1 | 18.4 | 55.8 | 28.3 | 40.4 |
| GPT-4o-mini | 11.5 | 44.9 | 48.5 | 30.3 | 21.8 | 36.4 | 24.1 | 58.6 | 8.16 | 6.12 | 12.2 | 56.5 | 19.6 | 43.0 |
| *w/ Traj.* | 12.8 | 44.9 | 39.4 | 25.8 | 27.3 | 36.4 | 20.7 | 48.3 | 12.2 | 12.2 | 14.3 | 57.1 | 19.9 | 41.9 |
| *w/ Guideline* | 12.8 | 43.6 | 47.0 | 37.9 | 27.3 | 30.9 | 31.0 | 51.7 | 12.2 | 4.10 | 12.9 | 60.5 | 21.9 | 43.5 |
| *w/ MetaFlow* | 19.2 | 30.8 | 53.0 | 15.2 | 16.4 | 23.6 | 31.0 | 46.6 | 16.3 | 4.08 | 22.4 | 51.7 | 26.1 | 33.6 |

construct an *experience pool* via rejection sampling from previously collected trajectories. The results are shown in Table 2, from which we draw three observations:

1. All three methods result in improvements in the model's overall performance. However, the MetaFlow method demonstrates the most significant enhancement. Specifically, when employing MetaFlow, the average increase in accuracy is **6.16**%, accompanied by a **4.13**% reduction in side effects. Furthermore, when using GPT-4o-mini as the backbone LLM, MetaFlow outperforms both trajectory-based and guideline-based experience methods across the six domains and evaluation metrics, with only one exception. This demonstrates the effectiveness of MetaFlowLLM in tool-calling tasks, as it can reuse experience more efficiently.

2. MetaFlow shows greater performance improvements with more powerful LLMs. For example, the accuracy of Qwen2.5-32B-Instruct improves by $9.1\%$, while Qwen2.5-7B-Instruct only increases by $2.9\%$, which is slightly lower than the performance of the guideline method at the same scale. Manual inspection reveals two key reasons: (1) low-capacity models exhibit insufficient discriminative capability to identify appropriate MetaFlows; (2) their weak instruction-following performance impedes the successful execution of MetaFlow sub-tasks.

## 4.3 Analysis

**Scalability Analysis.** Theoretically, both MetaFlowGen training and tree construction are performed offline, while online retrieval operates with sublinear time complexity relative to the number of nodes. The most computationally intensive operation—correctness reward computation—formally scales as $\mathcal{O}(N)$ with the number of leaf nodes but can be fully parallelized across GPUs. Empirically, the runtime of both RL training and tree construction grows approximately linearly with the number of experiences (see Figure 3).

**MetaFlow can reduce the impact of noisy context from more available tools. Selecting inappropriate MetaFlow nodes can hinder performance.** To investigate the significance of the discriminative model $\mathcal{P}(q|\mathcal{Q}, \mathcal{M})$, we evaluate MetaFlowLLM's performance under hierarchical level shifts. For each test task, we employ GPT-4o as the discriminative model to identify appropriate MetaFlow nodes. Subsequently, we evaluate performance deviations when accessing either parent nodes (with negative level shifts) or child nodes (with positive level shifts). As shown in Figure 4, both upward and downward shifts lead to degradation in model accuracy,

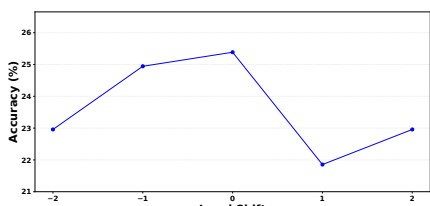

Figure 4: Comparison of accuracy with varying level shifts on WorkBench.

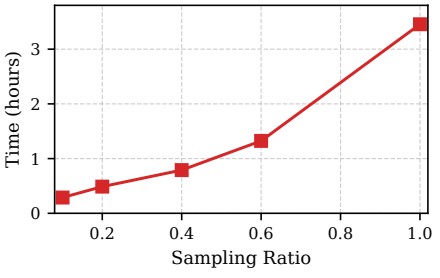

(a) Tree-building on *WorkBench*.

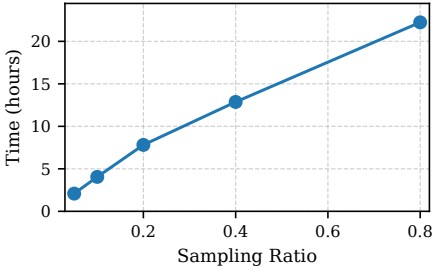

(b) RL training on *AppWorld*.

Figure 3: **Empirical runtime scaling of MetaFlowLLM.** Both curves exhibit approximately linear scaling as the sampling ratio increases.

indicating that the originally selected node is better than others. This demonstrates the importance of the discriminative model in the inference.

In our main experiments, we only provide the necessary tools to the ReAct agent. In this study, we evaluate the performance of MetaFlowLLM in a more real-world setting where all 26 tools are provided. As shown in Table 3, MetaFlowLLM results in a more considerable improvement in accuracy, while concurrently reducing side effects. For example, Qwen2.5-7B-Instruct achieves an accuracy improvement from 9.05% to 15.7%. This indicates that MetaFlowLLM mitigates the effect of noisy context by predefining tool usage in static steps and clearer planning in dynamic steps, thereby reducing the complexity of tool selection.

Table 3: Accuracy and side effects comparison with all tools on WorkBench (%).

| Model | Accuracy (%) | Side Effect (%) |
|---|---|---|
| Qwen2.5-7B | 9.05 | 47.9 |
| *w/ MetaFlow* | **15.7** | **41.7** |
| Qwen2.5-32B | 13.8 | 52.3 |
| *w/ MetaFlow* | **23.2** | **47.7** |

**The MetaFlowGen demonstrates superior performance over proprietary models.** To validate the effectiveness of the proposed training pipeline, we conduct comparative experiments involving GPT-4o, DeepSeek-R1, and our MetaFlowGen. We provided three in-context learning examples for GPT-4o and DeepSeek-R1, while the trained models utilize a zero-shot prompt. The experimental results (Table 4) reveal two findings: (1) The trained MetaFlowGen achieves significantly higher performance than both GPT-4o and DeepSeek-R1 across both

Table 4: Comparison between various MetaFlow generators on AppWorld.

| Model | Correct | Efficiency |
|---|---|---|
| GPT-4o | 0.160 | 0.60 |
| DeepSeek-R1 | 0.054 | 0.41 |
| MetaFlowGen | 0.360 | 0.99 |
| w/o RL | 0.160 | 0.80 |
| w/o SFT | -0.380 | -0.14 |

metrics; (2) Ablation studies demonstrate consistent performance degradation when either SFT or RL is removed, confirming the essential contribution of both proposed training stages. We also provide the reward curves during RL training and more detailed analysis in Appendix B.

**The proposed RL framework exhibits superior sample efficiency compared to the SFT baseline.** To evaluate the cost-effectiveness of our reinforcement learning pipeline, we vary the sampling ratio of the RL training set and analyzed the resulting correctness and efficiency metrics, as shown in Figure 5. We observe a consistent improvement in both metrics as the sampling ratio increases. Notably, the model attains near-optimal performance at a full sampling ratio (1.0), achieving a peak efficiency of 0.991. Although the correctness slightly declines at this point, the overall gain in efficiency highlights the inherent trade-off between accuracy and computational cost—an expected characteristic in RL optimization. These results demonstrate that the RL training method can achieve competitive performance with substantially fewer training samples, underscoring its strong sample efficiency.

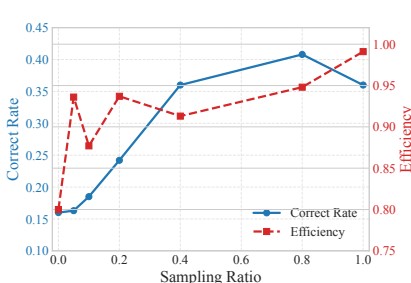

Figure 5: Sample efficiency of the proposed RL framework.

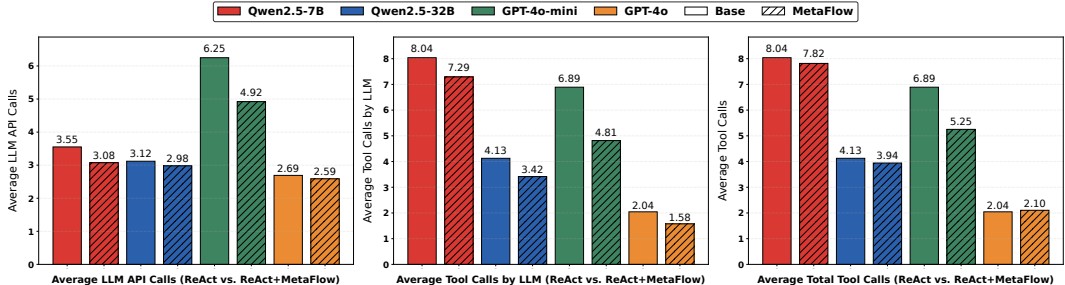

Figure 6: Efficiency comparison between ReAct and ReAct+MetaFlow, based on the average number of LLM API calls and tool calls.

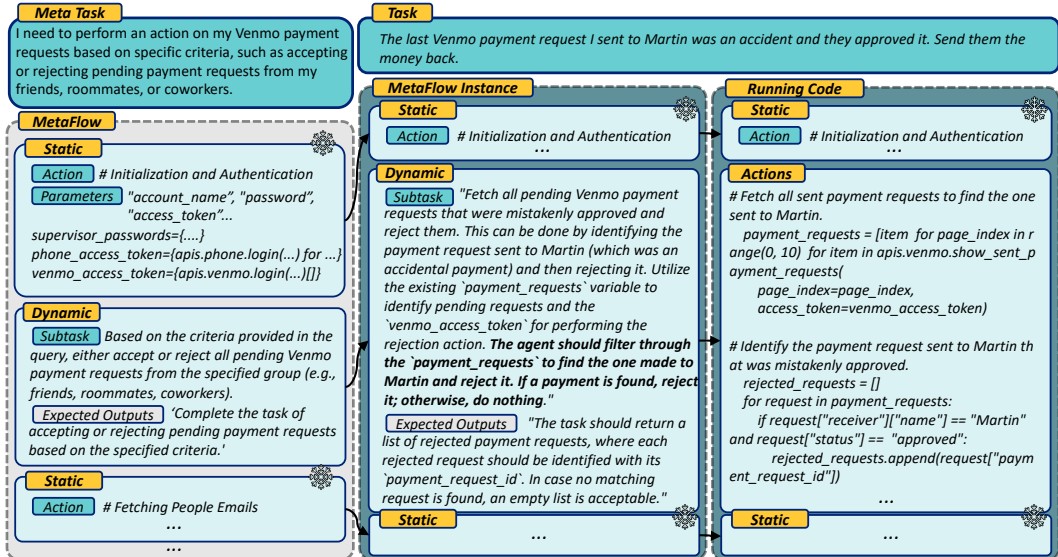

Figure 7: Case Study of Executing MetaFlows from AppWorld.

**MetaFlowLLM can enhance the efficiency of sequential decision-making.** We evaluate the performance of different models using ReAct and ReAct+MetaFlow based on three key metrics: the number of LLM API calls, the average number of tool calls by LLMs, and the average number of tool calls overall. As shown in Figure 6, all models demonstrate a consistent reduction in both the number of LLM API calls and the number of tool calls by LLMs when MetaFlow is applied. This indicates that the introduced static steps effectively address subproblems. In addition, we find that MetaFlowLLM helps reduce the overall number of tool calls, particularly when the base LLMs are relatively weak. Manual inspection indicates that the elevated tool call frequency is primarily due to the model's dependence on repeated trial and error during execution. More efficiency comparison results can be found in Appendix H.

**Case Study of Executing MetaFlows.** Figure 7 presents a MetaFlow execution example in AppWorld. The meta task and its corresponding MetaFlow are shown in the left part of the figure. Static steps are executed as-is, while dynamic steps are adapted according to the input task and context (transition from the left to the center part of Figure 7). The resulting subtasks are then used to prompt the agent, which generates and executes code accordingly (center to right part of Figure 7).

## 5 Conclusion

In this paper, we present MetaFlowLLM, a novel framework that enables structured experience reuse for LLM-based agents through a hierarchical experience tree. By abstracting past task trajectories into a tree of MetaFlows, our approach facilitates effective and efficient task execution for new-coming tasks. In future work, we plan to extend MetaFlowLLM to multimodal domains and beyond.

## Acknowledgements

This work was supported by the National Key R&D Program of China(No.2024YFC3306500), Beijing Nova Program (No. 20240484568), the Postdoctoral Fellowship Program of CPSF (Grant No. GZB20230343 and Grant No. GZC20240831) and the China Postdoctoral Science Foundation (Grant No. 2023M741945 and Grant No. 2025M771586).

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

## A Limitation and Broader Impacts

**Limitation.** While the framework proposed in this paper represents a notable advancement in LLM agents, it also has certain limitations that warrant discussion. First, the proposed framework requires offline data in order to organize hierarchical MetaFlows. Second, the rejection sampling method and reinforcement learning approach employed in this paper rely on verifiable rewards, which limits their applicability in some open-ended evaluation tasks. Therefore, further exploration of these methods' applicability in such tasks is required.

**Broader Impact.** This paper introduces research aimed at enhancing LLM agent capabilities through experience reuse. From a societal impact perspective, while we have developed a generic LLM-based autonomous agent, the presence of biased offline datasets may lead to decisions with suboptimal outcomes. Additionally, there is potential for autonomous agents to be misused in malicious applications. In response to these concerns, all data used in this paper is publicly available and does not involve private information. Moreover, the proposed framework should not be used for any malicious purposes.

## B Reward Dynamics During RL Training

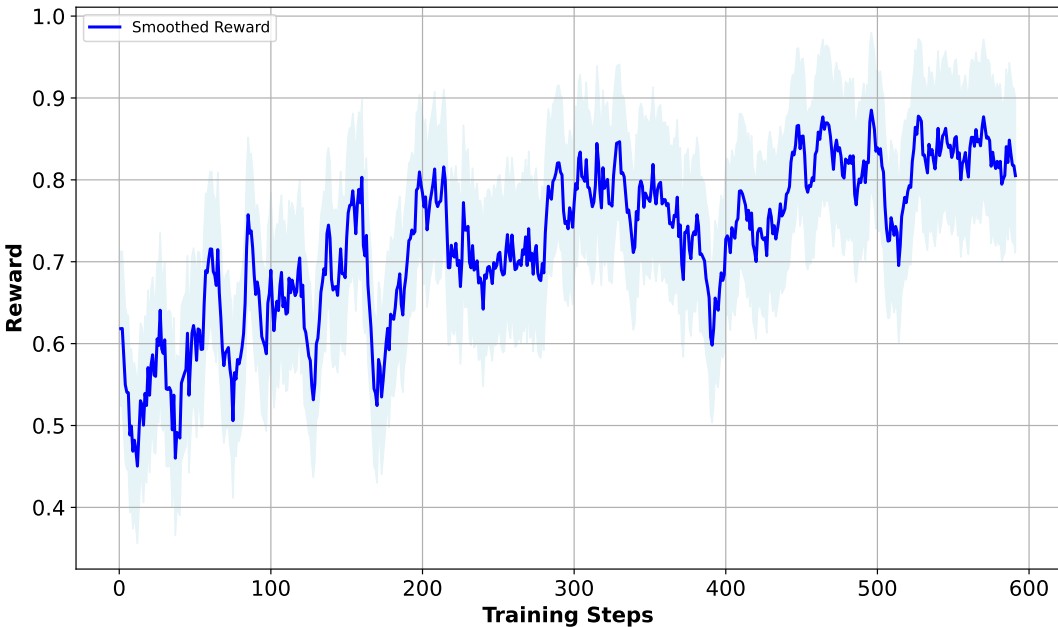

Figure 8: Reward Progression Across RL Training Steps.

As shown in Figure 8, despite variations in difficulty across different samples and the inherent uncertainty in the operation of the LLM agent leading to fluctuations in reward, the model's reward demonstrates a consistent improvement over the course of the three epochs in RL training.

Intuitively, during the RL process, the model generates various MetaFlows through rollouts. We calculate the reward by executing the MetaFlows at the leaf nodes, which is based on test case matching. This reward avoids the risk of reward hacking and is identical to the calculation process used during actual deployment. These rewards are then involved in the calculation of fitting weights for different trajectories. MetaFlows with higher rewards receive greater fitting weights, which steers the model's optimization towards these samples. On the other hand, MetaFlows with lower rewards are trained to avoid generating such samples.

For example, during the MetaFlowGen's rollout, two MetaFlows may be generated: one that does not meet the format specifications (resulting in a reward of -1) and another that adheres to the specifications, featuring more static steps and more detailed dynamic step instructions (resulting in a

---
**Algorithm 2** Deep Matching Traversal
---
**Require:** Hierarchical Experience Tree $\mathcal{T}$, task $q$, matching model $\mathcal{P}(q|\mathcal{Q}, \mathcal{M})$
**Ensure:** Best matching MetaFlow instance
 1: **function** FindBestMatch($node, q, depth$)
 2:   **if** $\mathcal{P}(q|\mathcal{Q}_{\text{node}}, \mathcal{M}_{\text{node}}) =$ False **then**
 3:     **return** None
 4: **end if**
 5: $best\_node \leftarrow node, max\_depth \leftarrow depth$
 6: **for** each child $c$ of $node$ **do**
 7:   $match \leftarrow$ FindBestMatch($c, q, depth + 1$)
 8:   **if** $match \neq$ None **then**
 9:     $child\_depth \leftarrow$ Depth of $match$
10:     **if** $child\_depth > max\_depth$ **then**
11:       $best\_node \leftarrow match, max\_depth \leftarrow child\_depth$
12:     **end if**
13:   **end if**
14: **end for**
15: **return** $best\_node$
16: **Start from root:** $best\_node \leftarrow$ FindBestMatch($\mathcal{T}_{\text{root}}, q, 0$)
17: **return** execute($q, \mathcal{M}_{best\_node}$)
---

reward of 1). The GRPO algorithm adjusts the probabilities accordingly—reducing the likelihood of generating non-compliant samples and increasing the probability of generating valid samples.

## C Deep Matching Traversal

Algorithm 2 illustrates the Deep Matching Traversal process, which recursively searches the hierarchical experience tree to find the deepest node whose experience best matches the query.

## D Dataset Statistics

For WorkBench, since the original repository only provided golden solutions without complete workflows, we first prompted Qwen-32B-Instruct and filtered responses that matched the golden solutions to obtain offline data. For AppWorld, we used the officially partitioned training set as the offline data. The dataset statistics are summarized in Table 5.

Table 5: Data statistics.

| Metric | WorkBench | AppWorld |
|---|---|---|
| Offline Data Size | 237 | 90 |
| Test Data Size | 353 | 57 |
| SFT Data Size | 1,102 | 1,092 |
| RL Data Size | 121 | 247 |

## E Implementation of Baseline Methods

The implementation details of the baseline methods are as follows.

**Trajectory-based baseline.** For each input task, we first encode its description using *all-MiniLM-L6-v2* to obtain a task embedding. Cosine similarity is then computed between this embedding and all task embeddings in the offline dataset. The most similar task is selected, and its description and solution trajectory are used as in-context learning (ICL) examples in the LLM agent's prompt.

**Guideline-based baseline.** We first prompt the same LLM to summarize guidelines from offline trajectories. For a given task, we again encode its description using *all-MiniLM-L6-v2* and find the most similar task via cosine similarity. The retrieved task's guideline and randomly selected ICL examples are then incorporated into the LLM agent's prompt.

The guideline summarization prompt is:

> Please summarize the experience in natural language from the following examples so that you can use this experience in subsequent similar tasks. Please don't rehash the code or discuss code details. `query: {query}` `solution: {trajectory}`

# F   Effect of Distance Metric on Experience Tree Quality

MetaFlowLLM is a general-purpose framework not tied to a specific distance metric. While our main experiments use task embeddings from *all-MiniLM-L6-v2* with cosine similarity, the framework also supports alternatives such as trajectory embeddings, Jaccard similarity, or rule-based measures.

As shown in Table 6, both trajectory- and task-based embeddings improve model performance over baselines. The slight gap between them likely arises from noise in trajectory representations caused by tool call responses.

Table 6: **Impact of Distance Metric in MetaFlowLLM.**

| Model | %Acc ↑ | %SE ↓ |
|---|---|---|
| Qwen2.5-7B | 14.3 | 43.0 |
| + Trajectory Embedding | 15.2 | **40.8** |
| + Task Embedding | **17.2** | 46.8 |
| Qwen2.5-32B | 19.2 | 47.2 |
| + Trajectory Embedding | 24.9 | 46.5 |
| + Task Embedding | **28.3** | **40.4** |
| GPT-4o-mini | 19.6 | 43.0 |
| + Trajectory Embedding | 25.1 | 37.1 |
| + Task Embedding | **26.1** | **33.6** |

# G   Case Study of MetaFlows

In this section, we present a typical MetaFlow example on WorkBench.

```
[
  {
    "type": "dynamic",
    "instruction": "Extract the comparison operator, threshold,
        and time period from the user query and calculate the
        start and end date. The time period is natural language
        like '1 week' or '2 weeks'. Output the start and end
        dates as ISO 8601 format.",
    "outputs": [
      "time_min",
      "time_max",
      "comparison_operator",
      "threshold"
    ]
  },
  {
    "type": "static",
    "action": "analytics.total_visits_count",
    "params": {
      "time_min": "${time_min}",
      "time_max": "${time_max}"
    },
    "output": "total_visits_data"
  },
```

```
{
    "type": "dynamic",
    "instruction": "Check if the total visits data meets the
        condition specified by the comparison operator and
        threshold. If so, generate a line chart using the
        analytics.create_plot function with the value_to_plot
        set to total_visits.",
    "outputs": [
      "plot_file_path"
    ]
  }
]
```

## H Inference Time Comparison

Table 7 presents the inference time comparison between the standard ReAct agent and the proposed ReAct+MetaFlowLLM framework on the WorkBench benchmark. From the results, we observe that incorporating MetaFlowLLM incurs only a minor increase in total inference time (approximately 9%) compared to the vanilla ReAct agent, while achieving a substantial improvement in accuracy. This demonstrates that MetaFlowLLM offers a favorable trade-off between efficiency and performance, making it a practical enhancement for real-world multi-step reasoning systems.

Table 7: **Inference time comparison on WorkBench.** Both methods use Qwen2.5-32B-Instruct as the underlying model.

| Method | Avg. Retrieval Time (s) | Avg. Running Time (s) | Accuracy (%) |
|---|---|---|---|
| ReAct | 0.0 | 12.1 | 19.2 |
| ReAct + MetaFlow | 1.7 | 11.5 | 28.3 |

## I Comparison with AWM

We conduct experiments using application type as the categories in AWM [34], and the experimental results on AppWorld are presented in the Table 8. These results demonstrate that AWM indeed provides improvements over the trajectory-based and guideline-based baselines in this paper. Notably, MetaFlowLLM consistently outperforms AWM across all backbones.

Table 8: **Performance comparison of various models and agent types on AppWorld (%).**

| Model | Agent Type | Configuration | | | | |
|---|---|---|---|---|---|---|
| | | Base | *w/ Traj.* | *w/ Guideline* | *w/ MetaFlow* | *w/ AWM* |
| Qwen2.5-7B | Reflexion | 7.6 | 32.7 | 13.5 | **37.4** | 35.6 |
| | ReAct | 6.4 | 37.4 | 17.0 | **43.9** | 40.4 |
| Qwen2.5-32B | Reflexion | 12.9 | 33.9 | 22.8 | **43.9** | 25.7 |
| | ReAct | 22.2 | 49.1 | 38.6 | **50.9** | 50.3 |
| GPT-4o-mini | Reflexion | 13.5 | 37.4 | 20.5 | **45.6** | 37.4 |
| | ReAct | 7.0 | 22.2 | 7.0 | **35.7** | 22.8 |

## J Prompt Design

The MetaFlow generation prompt was manually designed through iterative refinement during preliminary experiments, while agent prompts were adapted from the official prompts of their respective source benchmarks.

## J.1 MetaFlow Generation Prompt

SYSTEM:
You are tasked with analyzing two provided examples, each
    consisting of a user query and its corresponding workflow.
    Based on these examples, your objective is to generalize a
    meta workflow that can solve similar problems using a
    combination of static executable code and dynamic LLM agent
     configurations. The meta workflow you generalize should
    not only summarize the given samples, but it should be as
    general as possible so that it can solve similar problems
    by reusing static code and utilizing dynamic agents.
The meta workflow runs sequentially. When encountering static
    code, it is executed directly. When encountering a dynamic
    agent's JSON configuration file, an LLM agent is triggered
    to generate Python code to solve the specific task. The
    Python code generated by the LLM agent and the static code
    will operate within the same compilation environment,
    allowing them to reference and interact with each other.

# Key Requirements:

## Static Code:

The static code segments must be fully executable and should
    not contain any placeholders or dynamic interpolation logic
     (e.g., query-specific formatted strings).

These segments should perform reusable tasks that are common
    across similar problems.

Considering that the examples provided may vary significantly,
    making it impossible to reuse any static code, static code
    can be omitted. However, please avoid generating static
    code filled with comments, as this lacks practical
    significance. The explanatory text can be placed entirely
    in the configuration file of the dynamic agent.

## Dynamic Configurations:

The dynamic agent should complete task-specific tasks and be
    able to link the static code before and after, ensuring
    that they run without errors. The dynamic portions of the
    meta workflow should be defined in a JSON configuration
    file.

The JSON configuration must include:

- The `task_description` that clearly defines the query-
    specific logic using placeholders (e.g., {criteria}) to
    represent components of the user's query.
- The `expected_final_state` upon task completion, such as how
    variables are updated to enable integration with subsequent
     static code. For example, if a later static Python code
    requires variables to be defined during a dynamic agent's
    execution, this should be clearly indicated.

## Meta Query:

The meta query should describe the type of problem that the
    meta workflow can solve, without including all specific
    details from the input queries. It should use placeholders
    (e.g., [criteria]) to generalize the problem.

## Meta Workflow Structure:

The meta workflow should alternate between static code blocks
    and dynamic JSON configurations.

- Static code blocks must be self-contained and executable
    without modification. Code consisting entirely of comments
    is not acceptable.
- Dynamic JSON configurations should only contain placeholders
    for query-specific logic and should not interfere with the
    execution of static code.

USER:
**Now Summary the Following Examples:**
**API Documentation**:
{api_docs}

**Sample1**:
**Query**:
{query1}
**Workflow**:
{workflow1}

**Sample2**:
**Query**:
{query2}
**Workflow**:
{workflow2}

Output:

**Meta Query**:
[Provide the generalized meta query here]

**Meta Workflow**:
[Provide the generalized meta workflow, which should include
    the static code enclosed appropriately and a dynamic agent
    configuration in JSON format. The configuration should be
    clearly separated by `---` and enclosed within triple
    backticks (```) to maintain structure and clarity.]

## J.2 Agent Prompt Design in WorkBench

SYSTEM:
You are a step-by-step task-solving assistant. At each stage,
    you are only allowed to complete the current subtask based
    on the available variables, tools and instructions.
Today's date is Thursday, November 30, 2023 and the current
    time is 00:00.
Please remember the current date and time. Please use today's
    date for all dates unless otherwise specified. Meetings
    must not start before 9am or end after 6pm.
Please respond helpfully and accurately to the user.

All the information is in the given query or you can get it
    using the given tools. Please do not assume or will ask for
     my help.

USER:

You are solving a multi-step query within a pre-defined
    workflow. The full user query is:

{total_task}

And the pre-defined workflow is:
{workflow}

But you should now focus **only** on the following subtask:
{step['instruction']}

Here are the historical Logs prior to the completion of this
    sub_task:
{history_logs}
Use the current variables: {context}
Do not perform any part of the full task beyond this subtask.
    Do not call other tools or add outputs unless explicitly
    instructed.

## J.3  Agent Prompt in AppWorld.

SYSTEM:
You are a super intelligent AI Assistant whose job is to
    complete day-to-day tasks by writing code to interact with
    apps on behalf of your supervisor. Use API documentation to
     understand how to interact with the apps.

You will undertake a *multi-step conversation* using a python
    REPL environment. That is, you will write the python code
    and the environment will execute it and show you the result
    , based on which, you will write python code for the next
    step and so on, until you've achieved the goal. This
    environment will let you interact with app/s using their
    associated APIs on my behalf.

Currently, you need to address the final sub-task of a total
    task while ensuring the variables used are consistent with
    previous interactions. We have executed some history code
    in previous steps; that history code is provided here, and
    you may assume that any relevant state or variables from it
     are in memory. Proceed under the assumption that you can
    use or refer to prior variables as needed.

USER:
# App-wise API Documentation:
```yaml
{api_documentation_string}
```

ASSISTANT:
Understood.

USER:
You have access to the following imports already available in
    your coding environment.
```python
{available_imports}
```

These APIs should be called as python functions through the `
    apis` object. E.g., `apis.supervisor.show_profile(...)` for
     the `show_profile` API of the `supervisor` app. Note that
    you already have the `apis` object available in the
    execution environment, so do NOT to create it yourself.

You can also import anything from the Python *standard* library
     via code. However, you cannot use any systems-related
    packages, like os, shutils, etc, as they are disabled.

Let's say you want to obtain supervisors' app account passwords
     and get login access_token for one of their apps, let's
    say Spotify. You can write the following snippet as part of
     your code:

```python
# I should use supervisor.show_profile to get the supervisor's
    account name and password,
# then pass it to spotify.login API in terms of username and
    password
supervisor_profile = apis.supervisor.show_profile()
supervisor_passwords = __CURLY_OPEN__
    account_password["account_name"]: account_password["
        password"]
    for account_password in apis.supervisor.
        show_account_passwords()
__CURLY_CLOSE__
spotify_access_token = apis.spotify.login(
    username=supervisor_profile["email"],
    password=supervisor_passwords["spotify"],
)["access_token"]
# ... remaining code uses spotify_access_token variable as
    required.
```

Notice how the arguments passed to the APIs and outputs parsed
    from their outputs are as per the API documentation.

After you have completed the total task, you must call `apis.
    supervisor.complete_task`.
If the task is a question (e.g., "How many songs are in the
    Spotify queue?"), it must be called with an `answer`
    argument with an appropriate value. Use words or numbers
    only as answers, not full sentences, e.g., "10" in this
    case.

If the task is not a question, "Start my Spotify music player
    .", the `answer` argument should not be passed, or its
    values should be `None`.

ASSISTANT:
Got it.

```
USER:
# History Code and Feedback
{history}

# Total Instruction
{task_instruction}

# Subtask Instruction
{subtask_instruction}

Write the code to complete this subtask. Only generate valid
    Python code. It must be within markdown-styled ("```") code
     blocks.
Do NOT say or explain ANYTHING else.

# APIs allowed to Use
{{required_apis | join(", ")}}

Remember you:
- must only use APIs from the above list passing arguments and
    parsing outputs as per the provided documentation.
- must make all decisions autonomously on your own, and not
    wait or ask for anything external.
- must call `apis.supervisor.complete_task` at the end as per
    the above instruction.
- do not have access to any other packages except what is
    provided above and is part of the Python standard library.
- You shouldn't write all the code at once. Write small chunks
    of code and only one chunk of code in every step to make
    full use of the environment feedback. Make sure everything
    is working correctly before making any irreversible change.
- Before each time you write a specific code, you should think
    about the current situation in the form of a code comment
    and put it in the code blocks.
```

